# Immunogenicity and Safety of the Third Booster Dose with mRNA-1273 COVID-19 Vaccine after Receiving Two Doses of Inactivated or Viral Vector COVID-19 Vaccine

**DOI:** 10.3390/vaccines11030553

**Published:** 2023-02-27

**Authors:** Auchara Tangsathapornpong, Sira Nanthapisal, Kanassanan Pontan, Pornumpa Bunjoungmanee, Yamonbhorn Neamkul, Arthit Boonyarangkul, Supattra Wanpen, Waraphon Fukpho, Sumana Jitpokasem, Phuntila Tharabenjasin, Peera Jaru-Ampornpan

**Affiliations:** 1Department of Pediatrics, Faculty of Medicine, Thammasat University, Khlong Nueang, Khlong Luang, Pathum Thani 12120, Thailand; 2Research Unit in Infectious and Immunology, Faculty of Medicine, Thammasat University, Khlong Nueang, Khlong Luang, Pathum Thani 12120, Thailand; 3Clinical Research Center, Faculty of Medicine, Thammasat University, Khlong Nueang, Khlong Luang, Pathum Thani 12120, Thailand; 4Thammasat University Hospital, Thammasat University, Khlong Nueang, Khlong Luang, Pathum Thani 12120, Thailand; 5Department of Obstetrics and Gynecology, Faculty of Medicine, Thammasat University, Khlong Nueang, Khlong Luang, Pathum Thani 12120, Thailand; 6Chulabhorn International College of Medicine, Thammasat University, Khlong Nueang, Khlong Luang, Pathum Thani 12120, Thailand; 7Virology and Cell Technology Research Team, National Center for Genetic Engineering and Biotechnology (BIOTEC), Khlong Nueang, Khlong Luang, Pathum Thani 12120, Thailand

**Keywords:** COVID-19 vaccine, CoronaVac, AZD1222, mRNA-1273, booster dose, anti-SARS-CoV-2 IgG

## Abstract

The changes in the severe acute respiratory syndrome coronavirus 2 and the tapering of immunity after vaccination have propelled the need for a booster dose vaccine. We aim to evaluate B and T cell immunogenicity and reactogenicity of mRNA-1273 COVID-19 vaccine (100 µg) as a third booster dose after receiving either two doses of inactivated COVID-19 vaccine (CoronaVac) or two doses of viral vector vaccine (AZD1222) in adults not previously infected with COVID-19. The anti-receptor-binding-domain IgG (anti-RBD IgG), surrogate virus neutralization test (sVNT) against the Delta variant, and Interferon-Gamma (IFN-γ) level were measured at baseline, day (D)14 and D90 after vaccination. In D14 and D90, the geometric means of sVNT were significantly increased to 99.4% and 94.5% inhibition in CoronaVac, respectively, whereas AZD1222 showed inhibition of 99.1% and 93%, respectively. Anti-RBD IgG levels were 61,249 to 9235 AU/mL in CoronaVac and 38,777 to 5877 AU/mL in AZD1222 after D14 and D90 vaccination. Increasing median frequencies of S1-specific T cell response by IFN-γ concentration were also elevated in D14 and were not significantly different between CoronaVac (107.8–2035.4 mIU/mL) and AZD1222 (282.5–2001.2 mIU/mL). This study provides evidence for the high immunogenicity of the mRNA-1273 booster after two doses of CoronaVac or AZD1222 in the Thai population.

## 1. Introduction 

The coronavirus disease 2019 (COVID-19) pandemic, resulting in severe acute respiratory syndrome coronavirus 2 (SARS-CoV-2), has continued to affect the globe. The novel virus was initially identified from an outbreak in Wuhan, Hubei, China, in December 2019 [1]. Thailand was the first place to discover a case outside China in early January 2020 [2]. Most recently, there has been a large outbreak of COVID-19 in Thailand, reporting as of April 2022 that more than 3.9 million people were diagnosed with COVID-19, with more than 26,510 deaths [3]. Since May 2021, the Delta variant has been the culprit in outbreaks across many countries, including Thailand [4]. Vaccines against SARS-CoV-2 infection are recognized as the most encouraging way to reduce transmission of COVID-19. A range of COVID-19 vaccines is accessible worldwide. Importantly, the World Health Organization (WHO) maintains a current list of vaccine candidates that may be appended in the future [5].

In Thailand, vaccination of two doses of inactivated SARS-CoV-2 (CoronaVac, Sinovac Life Sciences, Beijing, China) or ChAdOx1 nCoV-19 (AZD1222, Oxford/AstraZeneca) were the standard COVID-19 vaccines available early in the pandemic (March 2021), while a small number of BNT162b2 (BioNTech/Pfizer), BBIBP (Sinopharm COVID-19 vaccine) and mRNA 1273 (Moderna COVID-19 vaccine) vaccines were subsequently procured in late 2021. The CoronaVac demonstrated effectiveness in safeguarding against severe infection and death, with a two-dose efficacy of 65.9% against SARS-CoV-2 infection and 86.3% against COVID-19–related death [6]. The randomized controlled trials for the vaccine efficacy of AZD1222 have been reported as 90% and 70.4% in low-dose priming and standard-dose groups, respectively [7]. However, the rate of vaccine efficiency experienced a gradual decline over the subsequent period, as affirmed by the rising incidence of symptomatic SARS-CoV-2 infection in vaccinated persons, as well as waning immunity [8,9]. Moreover, CoronaVac appeared to activate lower neutralizing antibodies against several variants [10].

New, emerging SARS-CoV-2 variants combined with declining immunity after vaccination have propelled the need for a booster vaccination. WHO has recommended that booster vaccination for COVID-19 may be a necessity, considering the tapering immunity, reduced vaccine effectiveness against several of the new variants, and the challenges of providing vaccine coverage on a worldwide scale [11]. A swift reduction of immunity against SARS-CoV-2 in the initial 3 months after receiving two-dose CoronaVac or AZD1222 highlights the need for a booster dose [9]. Currently, CoronaVac and AZD1222 have been widely used, and with the latest appearance of SARS-CoV-2 variants, a booster vaccination should be considered after the completion of the vaccination schedule.

The heterologous boosting strategy is defined as a vaccination different from the prior vaccine and could enhance the immunogenicity and broaden the B cell and T cell response against current worrisome SARS-CoV-2 variants [12,13]. Heterologous boosting of CoronaVac with a messenger RNA (mRNA) vaccine was demonstrated to enhance a stronger antibody response compared to other booster vaccine schedules [14,15,16]. Interestingly, it could generate neutralizing antibodies against ancestral, Delta, and other SARS-CoV-2 variants [14,15,16]. The Com-COV trial indicated that 28 days after being boosted, the geometric mean concentration of SARS-CoV-2 anti-spike IgG in heterologous AZD1222 followed by BNT162b2 was superior to two doses of AZD1222 [17]. Another study [12] and systematic review [18] also noted a strong immune response to this heterologous injection of AZD1222, followed by BNT162b2.

Nevertheless, there was no clear evidence to support the heterologous booster regimen in completed primary series vaccinated individuals. In Thailand, the recommendation regarding COVID-19 vaccination by the Ministry of Public Health stated that mRNA-1273 was an option for booster vaccines in individuals who completed primary vaccines.

Therefore, we hypothesized that the heterologous mRNA-1273 COVID-19 vaccine booster might improve the protection against COVID-19 variants. The mRNA-1273 vaccine is an effective mRNA vaccine for preventing symptomatic infection, hospitalization or death. The purpose of this study was to ascertain antibody response, cellular response, and reactogenicity of mRNA-1273 COVID-19 vaccine as the third booster dose after receiving two doses of CoronaVac or two doses of AZD1222 in healthy Thai adults.

## 2. Materials and Methods 

### 2.1. Study Design and Participants

This is a prospective observational study conducted at Thammasat University Hospital from November 2021–February 2022. A total of 100 adults aged 18 years and older who had previously received either two doses of CoronaVac for more than 30 days (N = 50) or the AZD1222 vaccine for more than 90 days (N = 50) were recruited. All activities/procedures were managed in accordance with the guidelines of the Human Research Ethics Committee of Thammasat University, Faculty of Medicine, Thammasat University (Institutional Review Board: approval No. MTU-EC-PE-6-255/64). The trial was registered with the Thai Clinical Trials Registry (registered number TCTwoR20211115003). Each participant was well informed of the protocol and possible undesirable effects, and signed informed consent was obtained prior to conducting any study activities/procedures. Subjects who had a history of COVID-19 infection or received any vaccines within 14 days prior to enrollment were excluded. All participants received a 100-μg single dose of the mRNA-1273 vaccine as the third booster vaccine. Blood samples were collected a total of 3 times in order to evaluate persistent immune reaction at baseline level (Day 0; D0), 2 weeks after vaccination (D14) and 3 months (D90). The serum IgG SARS-CoV-2 N protein was assessed at D0 to exclude asymptomatic COVID-19 infection. Twenty participants in the sub-study had the cell-mediated immune responses evaluated.

### 2.2. Study Procedures

Demographic and baseline characteristics (age, sex, height, and weight) and clinical data, including comorbidities, were screened to scrutinize inclusion/exclusion criteria. Participants received COVID-19 mRNA vaccine; mRNA1273-, manufactured by Moderna [19] (Lot number 30125BA and FH6387), 100-μg single dose intramuscularly at deltoid muscle, 0.5 mL. After vaccination, all participants were under observation for at least 30 min for any immediate post-immunization reactions. Diary cards were distributed to participants to record post-immunization local and systemic reactions (including pain/redness/swelling at the injection site, fever, headache, myalgia, arthralgia, fatigue, diarrhea, and vomiting) until Day 7 after immunization.

To evaluate the immune response to the booster vaccine, blood samples were taken from all participants at Visit 1 (Day 0) before vaccination (baseline), on Day 14 (Visit 2), and on Day 90 (Visit 3). Blood samples were further processed for serum separation at the study site laboratories. Serum samples were then shipped to designated laboratories to perform the immunogenicity assays and measured for Immunoglobulin G antibodies to the receptor binding domain (anti-RBD IgG) of the S1 subunit of the SARS-CoV-2 spike protein and surrogate viral neutralizing titer (sVNT) against B.167.2 (Delta variant). Participants with laboratory evidence of prior infection as a result of being positive for IgG SARS-CoV-2 N protein were excluded from the analysis.

The cell-mediated immunity (CMI) sub-study to assess T cell responses was conducted among a randomized cohort of 20 participants prior to vaccination and D14 via IFN-γ Release Assay (IGRA) assay.

### 2.3. Immunogenicity Outcomes

#### 2.3.1. The Receptor Binding Domain IgG (Anti-RBD IgG) and Anti-Nucleocapsid Protein (Anti-Np)

The humoral immune response to SARS-CoV-2 was assessed by binding domain antibody and measured anti-RBD IgG by Quant assay using ARCHITECT system (Abbott Diagnostics, Chicago, IL, USA). The results were expressed in AU/mL units. The results with an Anti-RBD level of more than 40,000 AU/mL were diluted within a limited range as per the manufacturer’s protocol. Anti-Np antibody was quantified by Elecsys Anti-SAR-CoV-2 (Roche, Basel, Switzerland), which is a combined anti-SARS-CoV-2 nucleocapsid protein IgA/IgG/IgM detection immunoassay. A level above 10 AU/mL was determined to be positive. 

#### 2.3.2. Surrogate Virus Neutralization Test (sVNT) 

Neutralizing antibody against SARS-CoV-2 was measured by sVNT in order to identify neutralizing antibodies to the Delta variant. The mechanism underpinning sVNT detects neutralizing antibodies targeting the receptor-binding domain of SARS-CoV-2 spike protein by ELISA [20], which was conducted at the National Center for Genetic Engineering and Biotechnology (BIOTEC), Thailand. Briefly, purified recombinant human ACE2 (hACE2) and the RBD of SARS-CoV-2 spike protein from the Delta variant were used. Diluted serum was mixed with conjugated horseradish peroxidase (HRP)-RBD. The mixture was subsequently incubated and added to 96-well plates coated with 0.1 µg of recombinant hACE2 ectodomain per well (GenScript). The neutralizing antibody level was determined and given as percent signal inhibition (% inhibition) using the following equation: % Inhibition = 100 × {1 − (sample OD450/negative OD 450)}

For the % inhibition of sVNT to Delta variant, ≥80% inhibition was defined as the cut-off level for the study.

#### 2.3.3. Interferon-Gamma (IFN-γ) Release Assay (IGRA) to Evaluate T Cell Responses

Detection of specific immunity toward SARS-CoV-2 after 16 h stimulation was performed via the SARS-CoV-2 IGRA stimulation tube and the IFN-γ ELISA assay as per the manufacturer’s instruction (Euroimmun, Lubeck, Germany). The results can be defined as follows: IFN-γ [SARS-CoV-2] - IFN-γ [blank] < 100 mIU/mL was deemed to be negative, 100–200 mIU/mL as borderline, and >200 mIU/mL interpreted as positive. The upper limit was 5000 mIU/mL.

### 2.4. Reactogenicity

Solicited local and systemic reactions were evaluated from Day 0 and Day 7 after immunization by a self-recorded diary card. The grading of reactions was classified according to the Guidance for Industry Toxicity Grading Scale for Healthy Adult and Adolescent Volunteers Enrolled in Preventive Vaccine Clinical Trials, 2007 [21]. A grading scale of 0 refers to no symptoms; grade 1 represents mild symptoms, which did not interfere with activities, or vomiting 1–2 times/day, or diarrhea 2–3 times/day; grade 2 described moderate symptoms, which interfered with activities or required medication to relieve the symptom, or vomiting more than two times/day or diarrhea 4–5 times/day; grade 3 was for severe symptoms, which were incapacitating, or diarrhea more than five times/day, and grade 4 depicted potentially life-threatening symptoms that necessitate hospitalization or emergency room visit. Fever was classified as grade 1 (38.0–38.4 °C), grade 2 (38.5–38.9 °C), grade 3 (39–40 °C), and grade 4 (more than 40 °C). Unsolicited adverse events were also documented by an investigator at every visit.

### 2.5. Statistical Analysis

The demographic, laboratory data and other continuous variables were reported by median (interquartile range [IQR]), while the categorical variables were reported by number and presented as a percentage. Statistical analyses of the differences in categorical and continuous variables between two groups were performed by chi-square, Fisher’s exact test and Wilcoxon rank sum test.

Two-sided 95%CIs was the antilogarithm of titers from the difference of two-independent *t*-tests. The adverse events were compared using a chi-square test. *p* values < 0.05 were considered to be statistically significant. All calculated *p*-values are two-sided. Statistical analysis was calculated using Stata version 15.1 (Stata Corp., College Station, TX, USA).

## 3. Results 

### 3.1. Baseline Characteristics

One hundred adults were initially enrolled in this study. Of the 100 subjects, three subjects in the CoronaVac group were excluded from the experiment because of positive anti-Np. During the follow-up period, four participants could not attend at D90, one participant was infected by COVID-19, one participant received another booster COVID vaccine-19 before D90, and two participants moved to another province, leading to ‘lost to follow-up’ (Figure 1).

The baseline characteristics of the participants and demographic data are shown in Table 1. The median interval from the second dose of CoronaVac to the mRNA-1273 booster dose was 143 days (IQR 138–168), and from the second dose of AZD1222 to the mRNA-1273 booster dose was 155 days (IQR 155–155). The median age of participants was 37 years (interquartile [IQR], 27–47), 42 years (28–49) in the CoronaVac group and 31 years old (27–46) in the AZD1222 group. The majority of the participants were female, at 67%; the AZD1222 primed group had a higher percentage of females than the CoronaVac primed group (46.8% in CoronaVac vs. 86% in AZD1222, *p* < 0.001). The median body mass index (BMI) of participants was 23.5 kg/m^2^ (IQR 21.7–29.3) in CoronaVac and 22.3 kg/m^2^ (IQR 19.6–24.3) in AZD1222. A total of 16 (16.5%) participants had comorbidities (19.2% in the CoronaVac group and 14.3% in the AZD1222 group) which were composed of hypertension (8.2%), diabetes mellitus (3.1%), chronic lung disease (3.1%), obesity (2.1%) and other conditions (3.2%).

### 3.2. Reactogenicity

Local and systemic adverse events in the first week after receiving the mRNA-1273 booster dose were recorded via a diary card. The majority of participants had mild adverse reactions after receiving the mRNA-1273 booster dose and resolved spontaneously. Pain at the injection site was the most common (88.7%), followed by myalgia (68%), fatigue (60.8%), headache (54.6%), swelling at the injected site (19.6%), redness at the injected site (9.3%), arthralgia (13.4%), fever (13%) diarrhea (6.2%) and vomiting (4.1%). Grade 3 or severe local and systemic reactions were reported in 7 subjects (7%).

The number of participants with local and systemic reactogenicities is shown in Figure 2 and Appendix A. CoronaVac-primed individuals experienced local and systemic reactogenicity similar to AZD1222-primed individuals (Figure 2). 

### 3.3. Immunogenicity

#### 3.3.1. SARS-CoV-2 Neutralizing Antibody by Surrogate Virus Neutralization Test

Compared to D0 at baseline, in 14 days post mRNA-1273 boosters, Geometric means (GMs) of sVNT to Delta variant significantly increased from 24.6% inhibition (95%CI 18.0–31.9) to 99.4% inhibition (95%CI 98.7–99.4) in CoronaVac and from 36.8% inhibition (95%CI 31.1–43.0) to 99.1% inhibition (95%CI 98.9–99.5) in AZD1222, as shown in Figure 3. After 90 days, the GMs of sVNT marginally decreased, with 94.5% inhibition (95%CI 91.4–97.6) in CoronaVac and 93.1% inhibition (95%CI 88.4–98.0) in AZD1222.

#### 3.3.2. Quantitative IgG against Receptor-Binding Domain

At D14 post-booster doses, the geometric mean titer (GMT) of anti-RBD IgG was significantly increased, from 160.9 AU/mL (95%CI 113.8–227.6) to 61,249 AU/mL (95%CI 49,352–76,014) in CoronaVac (*p* < 0.0001) and from 382.8 AU/mL (95%CI 297.8–492.2) to 38,777 AU/mL (95% CI 32,659–46,040) in AZD1222 (*p* < 0.0001), as shown in Figure 4. GMT of Anti-RBD IgG at D14 post booster in CoronaVac is significantly higher than AZD1222 (*p* = 0.0039).

At D90, the post-booster anti-RBD IgG in both groups was decreased. The GMTs of anti-RBD IgG decreased to 9235 AU/mL (95%CI 7527–11,330) in CoronaVac and 5877 AU/mL (95%CI 5008–6898) in AZD1222. GMT of Anti-RBD IgG in D90 in CoronaVac was significantly greater than AZD1222 (*p* = 0.0003). 

Compared to D0 at a baseline level, GMs of Anti-RBD-IgG fold rising at D14 in CoronaVac and AZD1222 groups were 380 (95%CI 262.1–552.7) and 101.3 (95%CI 77.96–131.6), respectively. GMs fold rising in CoronaVac was significantly higher than AZD1222 (*p* < 0.0001)

### 3.4. Cell-Mediated Immune Response by IGRA Assay

IFN-γ concentrations at baseline and D14 after the mRNA-1273 booster are shown in Figure 5. At baseline, three of ten subjects in the CoronaVac group and eight of ten subjects in the AZD1222 group had positive results for the IGRA assay (>200 mIU/mL). The median of the baseline IFN-γ concentration was 107.8 mIU/mL (IQR 38.5–307.7) and 282.5 mIU/mL (154.0–332.9) in the CoronaVac and AZD1222 groups, respectively. At D14 after the booster dose, all subjects in both groups had positive results, with the median at 2035 mIU/mL (IQR 1927–2109) in CoronaVac and 2001 mIU/mL (IQR 1867–2090) in the AZD1222 group. T cell responses were comparable between CoronaVac and AZD1222 groups in both D0 and D14.

## 4. Discussion

The present study detailed immunologic assessments in both humoral and cellular immune responses to the third booster dose of 100 ug mRNA-1273 COVID-19 vaccine. We documented anti-S-RBD IgG and sVNT to the Delta variant responsible for the outbreak during the study period in many countries, including Thailand. Our study revealed that previous CoronaVac-primed individuals had non-inferior immunogenicity responses to AZD1222-primed individuals measured by anti-RBD IgG and sVNT to the Delta variant. The results showed that mRNA-1273 booster vaccination was able to stimulate the immune responses after D14 to 100% neutralizing inhibition (titer ≥ 80 %) against SARS-CoV-2 Delta variant in both previous CoronaVac and AZD1222 vaccine and 91.4% of participants had neutralizing titers ≥80% inhibition against SARS- CoV-2 Delta variant until approximately 3 months after their booster.

Booster immunization with mRNA-1273 promotes high levels of anti-SARS-CoV-2 IgG antibody production, indicating that heterologous CoronaVac or AZD1222 and mRNA vaccination could strengthen the immune response to SARS-CoV-2. Moreover, the immune response in the population with CoronaVac/mRNA-1273 was better than the population with AZD1222/mRNA-1273. These results suggested a strong immune response can be induced in those who had previously received the inactivated vaccine with an mRNA-1273 booster. Although the mechanisms are relatively unknown, our results provide the priority order of heterologous CoronaVac and mRNA-1273. The results would be beneficial for vaccine management in the countries in which people are starting with the CoronaVac vaccination.

Our study found that the level of immunity at baseline was very low in both the CoronaVac and AZD1222 primed groups. Although previous clinical trial studies have documented the immunogenicity of two doses of CoronaVac [22,23] or two doses of AZD1222 [7], the antibody levels to safeguard against SARS-CoV-2 are forecast to diminish gradually. It was documented that immunity to SARS-CoV-2 experienced a steep decline in the first three months following two doses of the CoronaVac or AZD1222 vaccine [9]. At present, therefore, a booster dose against SARS-CoV-2 Delta and other variants is required to maintain a higher neutralizing antibody to prevent infection.

There are currently limited studies to compare immunogenicity between each licensed vaccine schedule. Moreover, immunity and reactogenicity after heterologous vaccination have garnered less attention. A previous study showed a strong antibody response to mRNA-1273 vaccine booster when given 6 months after the second CoronaVac vaccination, but T cell response data are lacking [24]. We uncovered that priming with the CoronaVac showed a comparable induction of T cell responses compared to AZD1222 priming, while antibody responses were more pronounced after CoronaVac priming. The beneficial effect of heterologous prime-boost immunizations with sequential administration of vaccines by using varying antigen delivery systems on enhancing the cellular immune response against a range of viral pathogens, including SARS-CoV-2, has been addressed [25,26]. The use of the lipid nanoparticle-containing mRNA BNT162b2 vaccine as a booster in participants having already received AZD1222 demonstrated significantly higher values of spike-specific CD4+ and CD8+ T cells compared with the two doses of AZD1222 [12].

Two groups of participants in our study had differences in gender distribution. The AZD1222-primed group had a higher percentage of females than the CoronaVac-primed group. However, the increases in anti-RBD IgG levels in both CoronaVac and AZD1222 groups are not different when normalized with sex, age or BMI.

Reports of solicited local and systemic reactions following mRNA-1273 vaccination were common manifestations [27]. Our result found higher rates of pain at the injection site, myalgia, fatigue, and headache in mRNA-1273 recipients. Nevertheless, both local and systemic reactions were mild. The adverse reactions in CoronaVac primed participants were similar to AZD1222 primed participants. Heterologous vaccine administration affected more frequent systemic reactions. Nevertheless, all reactogenicity could be tolerable and manageable [28].

The present study provides scientific evidence that a 100 μg booster dose of mRNA-1273 has an acceptable safety profile and generates a potent immune system in response to SARS-CoV-2 and other variants of concern. A previous study demonstrated that the 100 μg booster dose could enhance the comprehensiveness of the neutralizing antibody response and generate a higher titer of neutralizing antibody against different variants compared to a 50 μg booster dose. Despite having a reported clinically acceptable safety profile, both the regularity and severity of solicited adverse events with the 100 μg dose of the mRNA-1273 vaccine were greater than those with the 50 μg booster dose [29]. The underpinning mechanisms of the enhancement of antibodies in response to a 100 μg booster dose remain unknown at present. Further studies are needed for the process to be elucidated. Accordingly, the present study supported current evidence regarding heterologous vaccination that mRNA-1273 is a safe and effective option for the COVID-19 booster vaccine following the primary series with inactivated or viral vector COVID-19 vaccine.

Our study had certain limitations. First, we focused only on healthy adults and have no data in high-risk groups such as the immunocompromised or the elderly. Therefore, the current findings should be seriously considered while applying them to the patient. Second, this study did not measure neutralizing antibodies on recently emerged variants, e.g. Omicron. Third, a small number of sample sizes may be attributed to some non-significant outcomes. Studying a large population could be performed to strengthen our results.

Under the limitations, at least, the findings from this study can be applied to the public health policy for the recommendation of booster vaccine as mRNA-1273 vaccine that it can potentially induce strong immunogenicity in individuals who received a primary vaccine with either inactivated (CoronaVac or ChAdOx-1) or vector vaccine (AZD1222).

The conclusion of our study rests on the fact that the mRNA-1273 vaccine as a booster vaccination can efficiently enhance the SARS-CoV-2–specific B cell and T cell response following a prime dose of CoronaVac or AZD1222 vaccine, and it could strengthen prevention against SARS-CoV-2 infection, including the Delta variant.

## Figures and Tables

**Figure 1 vaccines-11-00553-f001:**
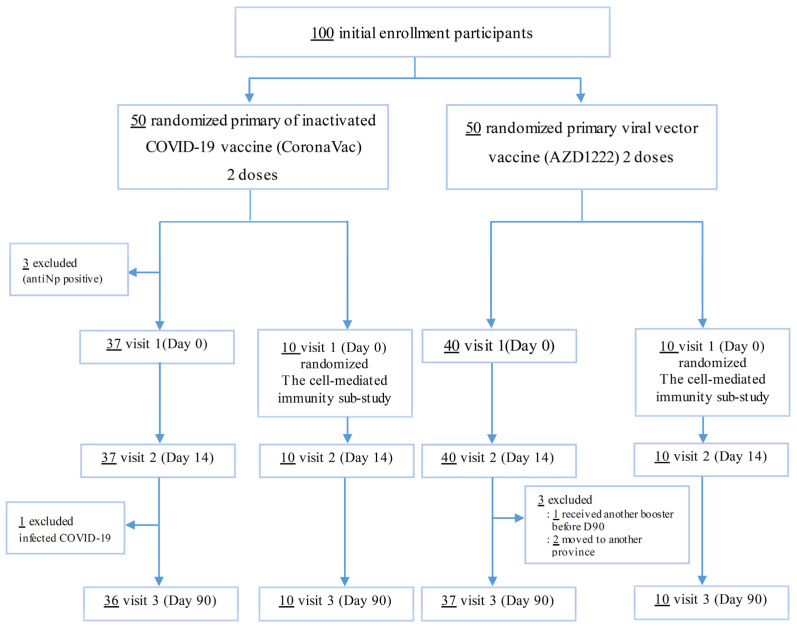
Flow diagram outlining the selection process for inclusion and exclusion of participations in the study.

**Figure 2 vaccines-11-00553-f002:**
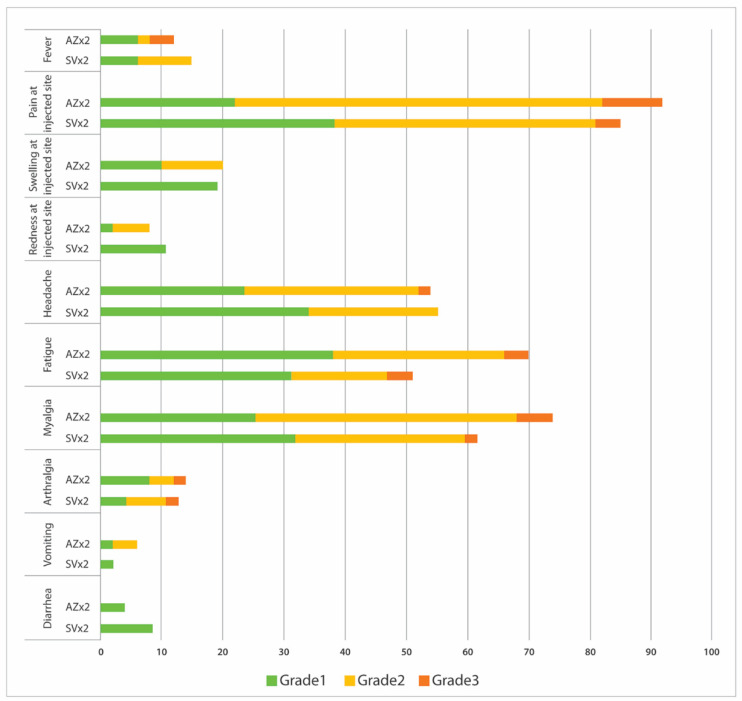
CoronaVac-primed individuals experienced local and systemic reactogenicity similar to AZD1222-primed individuals.

**Figure 3 vaccines-11-00553-f003:**
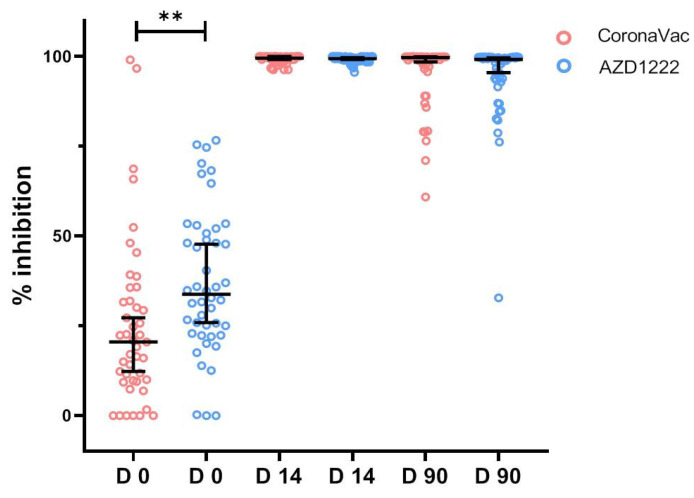
Surrogate virus neutralization test to Delta variants before, and D14 and D90.after booster vaccination with mRNA-1273 in participants who received two doses of CoronaVac and two doses of AZD1222. ** represented *p* value < 0.01.

**Figure 4 vaccines-11-00553-f004:**
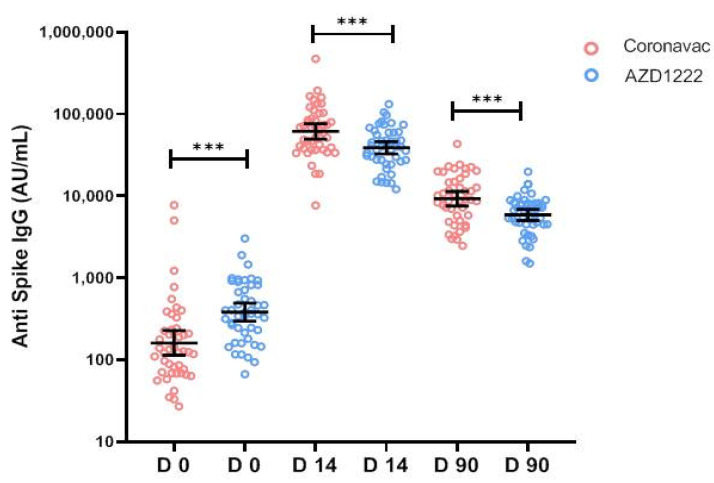
The geometric mean titer of anti-SARS-CoV2-receptor binding domain IgG before, D14 and D90 after booster vaccination with mRNA-1273 in participants who received two doses of CoronaVac and two doses of AZD1222. *** represented *p* value < 0.001.

**Figure 5 vaccines-11-00553-f005:**
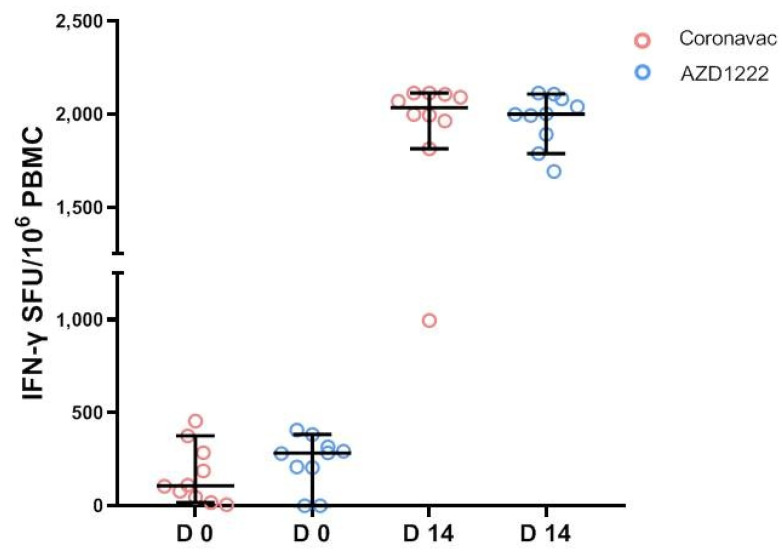
IFN-γ concentration before and day 14 after booster vaccination with mRNA-1273 in participants who received two doses of CoronaVac and two doses of AZD1222.

**Table 1 vaccines-11-00553-t001:** Baseline characteristics of the study participants.

Baseline Characteristic	Total (*n* = 97)	Post 2SV (*n* = 47)	Post 2AZ (*n* = 50)	*p*-Value
Age median y (IQR)	37 (27–47)	42 (28–49)	31 (27–46)	0.033
Female, *n* (%)	65 (67.01)	22 (46.81)	43 (86.00)	<0.001
BMI, median (IQR) <25 kg/m^2^, *n* (%) >25 kg/m^2^, *n* (%)	22.77 (20.5–25.39) 70 (72.16) 27 (27.84)	23.51 (21.72–29.32) 31 (65.96) 16 (34.04)	22.27 (19.61–24.24) 39 (78.00) 11 (22.00)	0.003
Comorbidities, *n* (%) Obesity *n* (%) Diabetes mellitus *n* (%) Cardiovascular disease *n* (%) Chronic lung disease *n* (%) Cancer *n* (%) Other *n* (%)	16 (16.49) 2 (2.06) 3 (3.09) 8 (8.25) 3 (3.09) 1 (1.03) 2 (2.06)	9 (19.15) 2 (4.26) 3 (6.38) 5 (10.64) 1 (2.13) 0 (0.00) 1 (2.13)	7 (14.00) 0 (0.00) 0 (0.00) 3 (6.00) 2 (4.00) 1 (2.00) 1 (2.00)	1.000 0.237 0.113 0.482 1.000 1.000 1.000
Interval between 2nd dose of CoronaVac or AZD1222 to mRNA1273 booster (day); median (IQR)	155 (143–157)	143 (138–168)	155 (155–155)	0.488
sVNT to Wuhan strain (%inhibition), GM (95%CI)	22.47 (19.67–25.66)	18.03 (14.62–22.25)	27.62 (23.75–32.12)	0.028
sVNT to Delta variant (%inhibition), GM (95%CI)	27 (22.86–32.62)	22.37 (17.44–28.69)	32.37 (25.21–41.57)	0.010
Anti-S-RBD IgG (BAU/mL), GM (95%CI)	36.96 (29.66–46.06)	23.49 (16.78–32.90)	56.59 (44.46–72.02)	0.637

AZ: AstraZeneca COVID-19 Vaccine; BMI: Body mass index; GM: Geometric mean; *n*: number of participants; S-RBD: Spike receptor binding domain; SV: Sinovac-CoronaVac vaccine; sVNT: Surrogate virus neutralization test.

## Data Availability

Data are contained within the article.

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
