# Peer review of "Immunogenicity and Safety of the Third Booster Dose with mRNA-1273 COVID-19 Vaccine after Receiving Two Doses of Inactivated or Viral Vector COVID-19 Vaccine"

_vaccines, 2023, doi:10.3390/vaccines11030553_

Round 1

Reviewer 1 Report

The manuscript entitled " Immunogenicity and safety of the third booster dose with mRNA COVID-19 vaccine after receiving two doses of inactivated or viral vector COVID-19 Vaccine" by Tangsathapornpong et al. is a straight forward study on the impact of a booster dose with mRNA 1273 Moderna COVID-19 vaccine to a selected group of individuals which have previously received two doses of either CoronaVac or AZD1222 vaccines. The results of this study indicated that there is not only a robust immunoglobulin G response to the 3rd dose but also high level of gamma interferon showing a strong T-cell activation as well. Therefore, I do not have any objections, if this manuscript is accepted for publication in Vaccines as a short communication.  

Author Response

Point 1: The manuscript entitled " Immunogenicity and safety of the third booster dose with mRNA COVID-19 vaccine after receiving two doses of inactivated or viral vector COVID-19 Vaccine" by Tangsathapornpong et al. is a straight forward study on the impact of a booster dose with mRNA 1273 Moderna COVID-19 vaccine to a selected group of individuals which have previously received two doses of either CoronaVac or AZD1222 vaccines. The results of this study indicated that there is not only a robust immunoglobulin G response to the 3rd dose but also high level of gamma interferon showing a strong T-cell activation as well. Therefore, I do not have any objections, if this manuscript is accepted for publication in Vaccines as a short communication.  

Response 1: We thank you very much to the reviewer for the support regarding our findings in this study that it would be a part of benefit to the other researcher as scientific information.

Reviewer 2 Report

The manuscript covers an interesting topic related to the immunogenicity and reactogenicity of mRNA 1273 COVID-19 vaccine as the third booster dose after receiving 2 doses of CoronaVac or 2 doses of AZD1222 in healthy Thai adults.

Please, add the sample size calculation.

Please, add the flow diagram of inclusion/exclusion participants.

please, add information regarding the setting.

in the discussion section, please add public health implications of your work.

Author Response

Point 1: Please, add the sample size calculation.

Response 1: We appreciate the reviewer for taking the time of reviewing our manuscript. Thank you very much for this suggestion.

Unfortunately, we did not calculate the sample size before starting our study. During the period of time, November 2021-March 2022, that we conducted this research, there were no previous evidences supporting the booster regimens particularly heterologous regimen for COVID-19 vaccination. Hence, we selected the number of participants per group to be 50 each based on an estimation and possibly recruited the subjects in that time. We could state that our investigation is a pilot study regarding specific of study period.

However, we added this issue according to the reviewer comment in the part of limitation of our study in the Discussion section. There might be the reason showing some of non-significant outcomes in our study.

Point 2: Please, add the flow diagram of inclusion/exclusion participants.

Response 2: Thank you very much for this excellent suggestion that we should simplify inclusion/exclusion participants by flow chart. Therefore, we inserted the flow chart in Figure 1 in the revised manuscript.

Point 3: Please, add information regarding the setting.

Response 3: Thank you very much for this suggestion.

By the time that the study was conducted, there was no clear evidence to support the heterologous booster regimen in completed primary series vaccinated individuals. In Thailand, the recommendation regarding COVID-19 vaccination by Ministry of Public Health stated mRNA-1273 was an option for booster vaccine in individuals who completed primary vaccines.

We added the below text in the revised manuscript in the section of Introduction.

“Nevertheless, there was no clear evidence to support the heterologous booster regimen in completed primary series vaccinated individuals. In Thailand, the recommendation regarding COVID-19 vaccination by Ministry of Public Health stated mRNA-1273 was an option for booster vaccine in individuals who completed primary vaccines.”

Point 4: In the discussion section, please add public health implications of your work.

Response 4: Thank you very much for this suggestion.

            We added “Under the limitations, at least, the findings from this study can be applied to the public health policy for the recommendation of booster vaccine as mRNA-1273 vaccine that it can potentially induce a strong immunogenicity in individuals who received primary vaccine with either inactivated (CoronaVac or ChAdOx-1) or vector vaccine (AZD1222)” into the latest part of discussion in the revised manuscript as public health implications of our work.

Reviewer 3 Report

Thank you for the opportunity of reviewing the ms. titled “Immunogenicity and safety of the third booster dose with mRNA COVID-19 vaccine after receiving two doses of Inactivated or Viral vector COVID-19 vaccine” by Tangsathapornpong and colleagues. 

The paper aims to investigate the response to booster dose of COVID-19 vaccination cycle with a mRNA COVID-19 vaccine, after mix-and-matched of doses. 

Overall, the paper is well written. A number of issues need revision tough. 

Vaccinees were administered the mRNA-1273 platform, but not BNT162b2: thus the title should be modified accordingly. This is due to the fact that the two mRNA platforms have  different response patterns in terms of safety (for post-third dose side effects, see: 10.3390/vaccines11020247) and immunogenicity (10.1126/scitranslmed.abm2311) , although not effectiveness (10.3390/vaccines10081353).

Authors might want to add some details what their paper adds to the available literature and to the current level of knowledge, also considering the small sample size of their study. 

Authors reported that vaccinees were administered with 100 μg booster dose of mRNA-1273. However, vials of mRNA-1273 vaccines for booster contain a lower dosage of mRNA 50 μg (compared with 100 of those used for primary vaccination): see: 10.1016/j.ijpharm.2021.120586 — Please clarify. 

Limitations of the study should be better acknowledged and not only limited to a cursory analysis of two points.  

Author Response

Thank you for the opportunity of reviewing the ms. titled “Immunogenicity and safety of the third booster dose with mRNA COVID-19 vaccine after receiving two doses of Inactivated or Viral vector COVID-19 vaccine” by Tangsathapornpong and colleagues. 

Point 1: The paper aims to investigate the response to booster dose of COVID-19 vaccination cycle with a mRNA COVID-19 vaccine, after mix-and-matched of doses. Overall, the paper is well written. A number of issues need revision tough. 

Response 1: We deeply thank you for the reviewer’s appreciation of our well written manuscript and taking your time for providing valuable comments and suggestions.

Point 2: Vaccinees were administered the mRNA-1273 platform, but not BNT162b2: thus the title should be modified accordingly. This is due to the fact that the two mRNA platforms have different response patterns in terms of safety (for post-third dose side effects, see: 10.3390/vaccines11020247) and immunogenicity (10.1126/scitranslmed.abm2311), although not effectiveness (10.3390/vaccines10081353).

Response 2: We are grateful to the reviewer for the suggestion regarding the title.

We reformulated the title of the revised manuscript by identifying “mRNA-1273 as third booster dose”.

Point 3: Authors might want to add some details what their paper adds to the available literature and to the current level of knowledge, also considering the small sample size of their study. 

Response 3: Thank you very much for the reviewer’s suggestion.

            We added details what our paper adds to the available literature and to the current level of knowledge in the Discussion section as follows “Accordingly, the present study supported current evidence regarding heterologous vaccination that mRNA-1273 is a safe and effective option to be used as COVID-19 booster vaccine following primary series with inactivated or viral vector COVID-19 vaccine”.

            To assuage the reviewer’s concern about sample size, we added the small sample size in part of the limitation in the Discussion section in the revised manuscript.

             As for the reason to select 50 participants in each group, we selected the number of participants based on an estimation and possibly recruited the subjects in that time. We could state that our investigation is a pilot study regarding specific of study period.

Point 4: Authors reported that vaccinees were administered with 100 μg booster dose of mRNA-1273. However, vials of mRNA-1273 vaccines for booster contain a lower dosage of mRNA 50 μg (compared with 100 of those used for primary vaccination): see: 10.1016/j.ijpharm.2021.120586 — Please clarify. 

Response 4: Thank you very much for the reviewer’s comment and consideration.

By the time we conducted this research, only mRNA-1273 vaccines in one dose 100 μg (0.5 mL) was under the conditional approved by the Thai authorities for human use in emergency situation during a pandemic crisis in Thailand. Moreover, there were no clear evidences indicating booster dosage of mRNA-1273 at 50 μg could be applied to individuals who received primary series vaccines with either inactivated (CoronaVac or ChAdOx-1) or vector vaccine (AZD1222). Additionally, the delta (Indian) variant of COVID-19 has becoming the dominant epidemic strain in Thailand and it was documented to resistant to the current vaccines. Taken together, we opted to use full booster dose of mRNA-1273 at 100 μg in our study.

Point 5: Limitations of the study should be better acknowledged and not only limited to a cursory analysis of two points.  

Response 5: Thank you very much for the reviewer’s comment and suggestion.

            In the revised manuscript, we more detailed about the limitation of each aspect. Furthermore, we added more one important of the limitation regarding small sample size.  

Round 2

Reviewer 3 Report

The paper can be accepted for publication